# IL-13Rα2 Regulates the IL-13/IFN-γ Balance during Innate Lymphoid Cell and Dendritic Cell Responses to Pox Viral Vector-Based Vaccination

**DOI:** 10.3390/vaccines9050440

**Published:** 2021-05-01

**Authors:** Zheyi Li, Sreeja Roy, Charani Ranasinghe

**Affiliations:** 1Molecular Mucosal Vaccine Immunology Group, Department of Immunology and infectious Disease, The John Curtin School of Medical Research, The Australian National University, Canberra, ACT 2601, Australia; zheyi.li@anu.edu.au (Z.L.); roys3@amc.edu (S.R.); 2Department of Immunology & Microbial Disease, Albany Medical College, 47 New Scotland Ave, Albany, NY 12208-3479, USA

**Keywords:** ILC, DC, IL-4R antagonist and IL-13Rα2 adjuvants, STAT6, IL-13, IL-4/IL-13 receptor regulation, viral vector vaccination

## Abstract

We have shown that manipulation of IL-13 and STAT6 signaling at the vaccination site can lead to different innate lymphoid cell (ILC)/dendritic cell (DC) recruitment, resulting in high avidity/poly-functional T cells and effective antibody differentiation. Here we show that permanent versus transient blockage of IL-13 and STAT6 at the vaccination site can lead to unique ILC-derived IL-13 and IFN-γ profiles, and differential IL-13Rα2, type I and II IL-4 receptor regulation on ILC. Specifically, STAT6^−/−^ BALB/c mice given fowl pox virus (FPV) expressing HIV antigens induced elevated ST2/IL-33R^+^ ILC2-derived IL-13 and reduced NKp46^+/−^ ILC1/ILC3-derived IFN-γ expression, whilst the opposite (reduced IL-13 and elevated IFN-γ expression) was observed during transient inhibition of STAT6 signaling in wild type BALB/c mice given FPV-HIV-IL-4R antagonist vaccination. Interestingly, disruption/inhibition of STAT6 signaling considerably impacted IL-13Rα2 expression by ST2/IL-33R^+^ ILC2 and NKp46^−^ ILC1/ILC3, unlike direct IL-13 inhibition. Consistently with our previous findings, this further indicated that inhibition of STAT6 most likely promoted IL-13 regulation via IL-13Rα2. Moreover, the elevated ST2/IL-33R^+^ IL-13Rα2^+^ lung ILC2, 24 h post FPV-HIV-IL-4R antagonist vaccination was also suggestive of an autocrine regulation of ILC2-derived IL-13 and IL-13Rα2, under certain conditions. Knowing that IL-13 can modulate IFN-γ expression, the elevated expression of IFN-γR on lung ST2/IL-33R^+^ ILC2 provoked the notion that there could also be inter-regulation of lung ILC2-derived IL-13 and NKp46^−^ ILC1/ILC3-derived IFN-γ via their respective receptors (IFN-γR and IL-13Rα2) at the lung mucosae early stages of vaccination. Intriguingly, under different IL-13 conditions differential regulation of IL-13/IL-13Rα2 on lung DC was also observed. Collectively these findings further substantiated that IL-13 is the master regulator of, not only DC, but also different ILC subsets at early stages of viral vector vaccination, and responsible for shaping the downstream adaptive immune outcomes. Thus, thoughtful selection of vaccine strategies/adjuvants that can manipulate IL-13Rα2, and STAT6 signaling at the ILC/DC level may prove useful in designing more efficacious vaccines against different/chronic pathogens.

## 1. Introduction

Cytokines IL-13 and IL-4 have been well studied in models that are related to Th2 immunity, such as allergy, asthma, parasitic, and helminth infections [1,2,3,4,5]. The roles of these two cytokines have been characterized as the regulators of Th1 and Th2 immune responses [4,5,6]. IL-4 and IL-13 signal via a common receptor system [7], where the type I and type II IL-4 receptor complexes consist of the γC/IL-4Rα and IL-4Rα/IL-13Rα1, respectively [8]. IL-4 binds to the type I receptor complex IL-4Rα with high affinity, and type II receptor complex with low affinity [9,10], activating the JAK/STAT6 pathway [11,12]. IL-13 can also bind to IL-13Rα1 of the type II IL-4 receptor complex, with low affinity (nM concentrations) and initiate signaling via the JAK/STAT6 pathway [8]. Furthermore, under low IL-13 conditions (pM concentrations), IL-13 is also thought to signal via the not well-characterized IL-13Rα2 pathway [13], involving STAT3 and activation of TGF-β1 [14]. Several studies have now shown that IL-13Rα2 can bind to IL-4Rα cytoplasmic tail and inhibit IL-4/IL-13 signaling via the IL-13Rα1/IL-4Rα type II complex and JAK/STAT6 [15,16,17,18]. Moreover, in cancer studies, IL-13Rα2 activation/signaling has been associated with TGF-β production in the absence of functional IL-4Rα [19]. Interestingly, dysregulation of IL-13Rα2 has been associated with many cancers [20,21,22,23].

The ILC are lineage negative cytokine-producing cells, which neither express lymphoid differentiation lineage markers nor T or B cell receptors. ILCs are generally divided into three distinct subsets, ILC1, ILC2, and ILC3, based on expression of cytokines, phenotypic markers, and transcription factors. Specifically, ILC2 are characterized by tissue specific surface expression of ST2/IL-33R (lung), IL-25R (muscle), or thymic stromal lymphopoietin receptor (TSLPR) (skin), and cytokines IL-4, IL-5, and IL-13, plus transcription factor GATA3 [24,25,26,27]. ILC1 and ILC3 are defined by the expression of NKp46, and their IFN-γ, IL-22, and IL-17A production capacity and linked to transcription factors T-bet and RORγt [27]. However, several studies have shown that the ILC populations can be highly plastic according to different cell/tissue milieus [24,25,28,29]. Recent studies in our laboratory have shown that following viral vector-based vaccination, ILC2 was the major source of IL-13 at the vaccination site 24 h post-delivery [24,30]. This was also linked to differential DC recruitment and downstream adaptive immune outcomes, which were also route dependent [24,31,32,33,34,35,36], suggesting that the optimal balance/regulation of IL-13 at the first line of defense was likely crucial for cell homeostasis and immune regulation.

Recently, we have designed two poxviral vector-based HIV vaccines that transiently manipulate IL-13 and IL-4 activity at the vaccination site to improve vaccine efficacy [32,33]. The IL-13Rα2 adjuvanted vaccine co-expressed HIV antigens together with soluble IL-13Rα2, which transiently sequestered IL-13 at the vaccination site [24,32]. The IL-4R antagonist adjuvanted vaccine co-expressed HIV antigens together with C-terminal deletion mutant of the mouse IL-4, lacking the essential tyrosine required for signaling. This antagonist was able to bind to both type I and type II IL-4 receptor complexes and transiently block both IL-4 and IL-13 signaling via the STAT6 pathway [33]. In a prime-boost modality, these vaccines were able to induce high avidity/poly-functional HIV specific mucosal and systemic CD4/CD8 T cells with improved protective efficacy, both in mice and macaques [31,32,33,34,35]. Moreover, in the context of humoral immunity, the IL-4R antagonist adjuvant vaccine was able to induce effective HIV gag-specific IgG1 and IgG2a differentiation in mice, unlike IL-13Rα2 adjuvanted vaccine [33]. To further confirm the role of IL-13 in antibody differentiation, when a cohort of knockout mice were vaccinated with unadjuvanted prime-boost strategy, although IL-4^−/−^ and STAT6^−/−^ animals showed enhanced IgG2a, IL-13^−/−^ mice showed extremely low IgG2a antibody responses [37]. More interestingly, when STAT6^−/−^ mice were given the IL-13Rα2 adjuvanted vaccine, elevated IgG1 and low IgG2a antibody responses were observed, similarly to the IL-13^−/−^ mice given the unadjuvanted vaccine [37]. These observations clearly indicated that the presence of IL-13 at the vaccination site was critical for effective antibody differentiation, and an STAT6 independent pathway was involved in this process, likely associated with IL-13Rα2 [37]. Interestingly, IFN-γ is also known to play an important role in antibody differentiation [38,39,40,41,42]. Moreover, during inflammation, IFN-γ has shown to inhibit ILC2 activation and IL-5 production [43], and similarly, under airway hyperreaction and asthma conditions IFN-γ has also shown to directly inhibit ILC2 function [44,45]. Recently we have also shown an interesting association of IL-13Rα2 and IFN-γR with different DC subsets under different IL-13 conditions [14]. However, the relationship between IL-13 and IFN-γ at the ILC/DC level in the context of viral vector-based vaccination, and the molecular mechanism by which ILC2-derived IL-13 regulates ILC1/ILC3 or DC activity, remain elusive. 

Therefore, in this study wild type (WT) BALB/c, IL-13, and STAT6 gene knockout mice on BALB/C background were vaccinated with unadjuvanted (FPV-HIV) and WT BALB/c mice with IL-13Rα2 or IL-4R antagonist adjuvanted viral vector-based vaccines (two models which cause enduring and transient inhibition of IL-4/IL-13/STAT6, respectively), to unravel the IL-4/IL-13 receptor regulation mechanisms on ILC and DC under different IL-13 conditions. 

## 2. Materials and Methods

Mice: 5–6 week old female wild type (WT) BALB/c, IL-13^−/−^, and STAT6^−/−^ mice on BALB/c background were obtained from the Australian Phenomics Facility, the Australian National University. 

Ethics Statement: All animals were maintained and experiments performed in accordance with the Australian National Health and Medical Research Council (NHMRC) guidelines, within the Australian Code of Practice for the Care and Use of Animals for Scientific Purposes. The animal ethics were approved by The Australian National University’s Animal Experimentation and Ethics Committee (AEEC). Protocol numbers A2014/14, and A2017/15.

Immunization: ILC studies, WT BALB/c mice were immunized with unadjuvanted FPV-HIV vaccine as control. IL-13^−/−^ and STAT6^−/−^ mice were also given the unadjuvanted FPV-HIV vaccine (which represented permanent IL-13 and STAT6 inhibition conditions, respectively) [36]. Another set of WT BALB/c mice were also immunized with FPV-HIV-IL-13Rα2 or FPV-HIV-IL-4R antagonist adjuvanted vaccines (which represented transient inhibition of IL-13 or IL-4/IL-13 signaling via STAT6, respectively) [7,32,33,34,35,46,47]. DC studies, FPV-HIV (rFPV) or VV-HIV (rVV) were administered to WT BALB/c mice. Each vaccine 10^7^ PFU was administered intranasally (i.n.) to mice (n = 4 to 6) under mild isoflurane anesthesia. Vaccines were diluted in sterile PBS, sonicated 3 times (15 s each time) on ice at 50 outputs using a Branson Sonifier 450 prior to administration, and given 10–15 µL per nostril (total 25–30 µL volume). Note that: (i) all vaccines were prepared as described previously [32,33,48]; (ii) rFPV is a non-replicating vector, and all co-expressed antigens/adjuvants were expressed for less than 72 h (transiently), which was also established by imaging studies [49,50], unlike rVV, which is a replicating vector that can express for much longer periods. 

Preparation of lung lymphocytes: The mice were euthanized using cervical dislocation according to the approved AEEC guidelines. Lung tissues were removed and kept in complete RPMI medium (Sigma) on ice until processing. Single cell lung suspensions were prepared as described previously [24,32]. Specifically, the lung tissues were first cut into small pieces, and then enzymatically digested in 1 mL of digestion buffer containing 1 mg/mL collagenase (Sigma-Aldrich, St Louis, MO, USA), 1.2 mg/mL Dispase (Gibco, Auckland, New Zealand), and 5 Units/mL DNase (Calbiochem, La Jolla, CA, USA) in complete RPMI. During digestion, samples were gently vortexed every 10 min and incubated in a 37 °C water bath for 45 min. The digested lung tissues were mashed and passed through a 100 µm Falcon cell strainer and the resulting lung cell suspensions were centrifuged for 15 min at 1500 RPM (524× *g*) at 4 °C using a Beckman ALLEGRA X-12R centrifuge. Next, the supernatants were removed, cells were resuspended in 5 mL red blood cell lysis buffer (at room temperature) containing 0.16 mM NH4Cl and 0.17 M Tris HCl (pH 7.65) for 3 min at room temperature, 30 mL of complete RPMI medium was added, and centrifuged at 1500 RPM (524× *g*) for 5 min at 4 °C. Cells were washed once more with complete RPMI and passed through sterile gauze to remove any remaining debris, followed by two washes in complete RPMI medium, and the cell pellets were then resuspended in 0.5 mL complete RPMI medium, counted using a hemocytometer (Tiefe Depth Profondeur 0.100 mm), and stored on ice until use.

Flow cytometry staining of ILC subsets and IL-4/IL-13 receptors: 2 × 10^6^ cells per sample were plated into U-bottomed 96-well plates (Falcon) and rested for 16 h at 37 °C with 5% CO_2_ in a Forma Scientific water-jacketed incubator, to allow the recovery of cell surface markers before performing the surface and intracellular staining [32]. Prior to staining, 1× Brefeldin A (BFA) was added to each sample and incubated at 37 °C with 5% CO_2_ for 5 h to prevent cytokine release. Surface and intracellular staining were performed as described previously [24]. Basically, cells were washed twice with FACS buffer (2% FCS in PBS), and FC block antibody (PharMingen clone 2.4G2) was added to reduce non-specific binding. Then cells were washed, and surface staining was performed for 40 min with the respective antibodies, including the IL-4/Il-13 receptors. Cells were next washed, fixed with IC-FIX buffer (Biolegend, San Diego, CA, USA), and permeabilized using IC-PERM buffer (Biolegend). Intracellular antibodies were then added for 30 min, washed, and cells were fixed with 0.5% PFA. Specifically, ILC2 staining; APC/Cy7-conjugated anti-mouse CD45 (Biolegend clone 30-F11) and FITC-conjugated lineage cocktail (CD3 (Biolegend clone 17A2), CD19 (Biolegend clone 6D5), CD11b (Biolegend clone M1/70), CD11c (Biolegend clone N418), CD49b (Biolegend clone HMα2), FcεRI (Biolegend clone MAR1)) were used to gate out the lineage-cells. PE-conjugated or PerCP/Cy5.5-conjugated anti-mouse ST2/IL-33R (Biolegend clone DIH9) was used to identify the lung ILC2. PE-eFlour 610-conjugated anti-mouse IL-13 (eBioscience clone EBio13A) was used to evaluate intracellular expression of IL-13 in ILC2s. For ILC1/ILC3 staining; APC/Cy7-conjugated anti-mouse CD45 (Biolegend clone 30-F11), and the same FITC-conjugated lineage cocktail were used to identify lineage- cells. PE-conjugated or PerCP/Cy5.5-conjugated anti-mouse ST2/IL-33R (Biolegend clone DIH9), and Brilliant Violet 421-conjugated anti-mouse CD335 (NKp46) (Biolegend clone 29A1.4) were used to identify ILC1/3 populations. Brilliant Violet 510-conjugated anti-mouse IFN-γ (Biolegend XMG1.2) was used to evaluate intracellular expression of IFN-γ in ILC1 and ILC3 and intracellular staining was performed as for ILC2.

PE-conjugated anti-mouse γC (CD132) (PharMingen clone 554457), PE-conjugated anti-mouse IL-4Rα (CD124) (Biolegend clone I015F8), PE-conjugated anti-mouse IL-13Rα1 (eBioscience clone 13MOKA), and Biotin-conjugated anti-mouse IL-13Rα2 (R&D, clone BAF539) were used to evaluated type I (γC and IL-4Rα) and type II (IL-4Rα and IL-13Rα1) IL-4 receptor complex and IL-13Rα2 expression on different ILC subsets. PE-conjugated streptavidin (Biolegend) was used as a secondary antibody to detect the Biotin-conjugated IL-13Rα2 antibody. All receptor antibodies were stained separately, to avoid spectral overlap. Specifically, γC (PE), IL-4Rα (PE), IL-13Rα1 (PE), and IL-13Rα2 (Biotin) were stained with ILC master mix antibodies. Then γC, IL-4Rα, and IL-13Rα1-stained samples were directly fixed with 0.5% PFA. The IL-13Rα2-stained samples were washed once with FACS buffer and then stained with PE-conjugated Streptavidin for 15 min on ice in the dark, followed by washing and fixing with 0.5% PFA. A total of 1,000,000 events were acquired per sample and analyzed using a BD LSR Fortessa. All ILC subsets and their γC, IL-4Rα, IL-13Rα1, and IL-13Rα2 expression were analyzed based on the fluorescent minus one (FMO) control, as described in Appendix A.

Evaluation of IL-4 and IL-13 receptor expression on lung cDCs and pDCs using flow cytometry: Lung tissues were harvested and prepared into single cell suspensions as for the ILC studies, 24, 48, or 72 h post vaccination. Then, 2 × 10^6^ cells from each sample were blocked with anti-mouse CD16/CD32 Fc Block antibody (BD Biosciences, San Jose, CA, USA) for 20 min at 4 °C, and cells were stained with DC markers, APC-conjugated anti-mouse MHCII I-Ad (e-Biosciences, San Diego, CA, USA), biotin-conjugated anti-mouse CD11c (N418 clone, Biolegend, San Diego, CA, USA), followed by streptavidin Brilliant violet 421 (Biolegend, San Diego, CA, USA), anti-mouse CD11b AlexaFluor 700 (M1170 clone, Biolegend, San Diego, CA, USA), anti-mouse CD103 FITC (2E7 clone, eBiosciences, San Diego, CA, USA), and anti-mouse B220 PercpCy5.5 (RA3-6B2 clone, e-Biosciences, San Diego, CA, USA) for 30 min on ice. To evaluate IL-4 and IL-13 receptors, cells were also extracellularly stained with either anti-mouse IL-4Rα (CD124) PE (I015F8 clone, Biolegend, San Diego, CA, USA), anti-mouse IL-13Rα1 (CD213a) PE (13MOKA clone, eBiosciences, San Diego, CA, USA), or Biotin-conjugated anti-mouse IL-13Rα2 (110815 clone, R&D systems, Minneapolis, MN, USA), followed by streptavidin PE (Biolegend, San Diego, CA, USA), anti-mouse γC (CD132) PE (TUGm2 clone, Biolegend, San Diego, CA, USA). Cells were fixed using 1.5% paraformaldehyde, resuspended in PBS and analyzed using a BD LSRII flow cytometer (Becton Dickinson, San Diego, CA, USA). A total of 5 × 10^5^ events per sample were acquired and the results were analyzed using FlowJo software v10.0.7 and the gating strategies described in Appendix A.

Statistical analysis: In this study, cell numbers were calculated using the formula (number of receptor or cytokine expressing cells/number of CD45^+^ cells) × 10^6^. Results are represented as a percentage of CD45^+^ cells, and can also be found in the Appendix A. IL-4 and IL-13 receptor proportions on lung DCs were calculated as a percentage of parent MHC-II^+^ CD11c^+^ CD11b^+^ CD103^−^ cDC and MHC-II^+^ CD11c^+^ CD11b^−^ B220^+^ pDC population. Note that less than 10 cells expressing the receptor was set as a cut-off. Statistical analysis was performed using GraphPad Prism software (version 6.05 for Windows). One-way ANOVA using Tukey’s multiple comparisons test and unpaired *t*-test were used. The *p*-values are denoted as: ns—*p* ≥ 0.05, *—*p* < 0.05, **—*p* < 0.01. ***—*p* < 0.001, ****—*p* < 0.0001. All experiments were repeated at least three times.

## 3. Results

### 3.1. Following rFPV Vaccination; ILC2-Derived IL-13 and ILC1/ILC3-Derived IFN-γ Expression Was Inversely Correlated

Mice were immunized with the unadjuvanted FPV-HIV, FPV-HIV-IL-4R antagonist, or FPV-HIV-IL-13Rα2 adjuvanted vaccines as described in the materials and methods. Then, 24 h post vaccination, IL-33R/ST2^+^ ILC2-derived IL-13 and NKp46^+/−^ ILC1/ILC3-derived IFN-γ expression profiles were evaluated using multicolor flow cytometry and gating strategy, indicated as described previously [24] (Appendix A). In this study the STAT6^−/−^ mice given the unadjuvanted vaccine showed the highest IL-33R/ST2^+^ ILC2-derived IL-13 expression and the lowest NKp46^+/−^ ILC1/ILC3-derived IFN-γ expression compared to all the other vaccine groups tested (Figure 1a–c). Interestingly, an inverse relationship was observed when WT BALB/c mice were given the FPV-HIV-IL-4R antagonist vaccination (Transient inhibition of STAT6 signaling) (Figure 1a–c). The IL-13^−/−^ mice given the unadjuvanted vaccine showed much greater IFN-γ expression by both NKp46^+/−^ ILC1/ILC3 compared to the WT BALB/c mice given the FPV-HIV-IL-13Rα2 adjuvanted vaccination (transient inhibition of IL-13) (*p* < 0.01) (Figure 1b,c). Between the different vaccination conditions tested the IL-33R/ST2^+^ ILC2-derived IL-13 expression profile was in the order: STAT6^−/−^ unadjuvanted > WT unadjuvanted > FPV-HIV-IL-4R antagonist or FPV-HIV-IL-13Rα2 adjuvanted > IL-13^−/−^ unadjuvanted vaccination. Whereas, the ILC1/ILC3-derived IFN-γ expression profile was WT unadjuvanted and FPV-HIV-IL-4R antagonist adjuvanted > IL-13^−/−^ unadjuvanted > FPV-HIV-IL-13Rα2 adjuvanted > STAT6^−/−^ unadjuvanted vaccination. No ILC2-derived IL-4 expression was detected in any of the vaccine groups tested. (For data presented in the form of percentage of CD45^+^ cells, please see Appendix A).

### 3.2. Elevated Number of ST2/IL-33R^+^ ILC2 and NKp46^−^ ILC1/ILC3 Expressed IL-13Rα2 Ollowing FPV-HIV-IL-4R Antagonist Vaccination, Unlike STAT6^−/−^ Given FPV-HIV

To examine type I and type II IL-4 receptor complexes and IL-13Rα2 expression profiles on different ILC subsets, STAT6^−/−^ and WT BALB/c mice were vaccinated intranasally with the unadjuvanted or adjuvanted rFPV vaccines. ILC profiles were evaluated 24 h post vaccination using the multicolor flow cytometry and gating strategy described in the materials and methods (Appendix A). Interestingly, vaccination under transient versus permanent STAT6 inhibition showed significantly different IL-13Rα2 and IL-4Rα type I receptor (IL-4Rα/γC) expression profiles on lung ST2/IL-33R^+^ ILC2. Specifically, an elevated number of ST2/IL-33R^+^ ILC2 were found to express IL-13Rα2 following IL-4R antagonist adjuvanted vaccination compared to WT BALB/c mice given the unadjuvanted FPV-HIV vaccination (*p* < 0.05) (Figure 2a), whereas no expression of IL-13Rα2 was observed in ST2/IL-33R^+^ ILC2 obtained from STAT6^−/−^ mice given the unadjuvanted vaccine (Figure 2a). Moreover, although no differences in IL-4Rα and γC expression were detected in ST2/IL-33R^+^ ILC2 obtained from WT BALB/c mice vaccinated with IL-4R antagonist adjuvanted and unadjuvanted vaccines, STAT6^−/−^ mice given unadjuvanted FPV-HIV vaccine showed significantly lower expression of these two receptors (Figure 2c,d). Interestingly, IL-13Rα1 expression on ST2/IL-33R^+^ ILC2 was not significantly different in the three vaccination groups tested (Figure 2b).

Next, when the densities (mean fluorescence intensities) of these receptors were accessed, no significant differences in IL-13Rα2 and IL-13Rα1 were detected on ST2/IL-33R^+^ ILC2s (Appendix A). However, IL-4Rα densities on ST2/IL-33R^+^ ILC2 obtained from STAT6^−/−^ mice given the unadjuvanted vaccine were significantly down-regulated compared to WT BALB/c mice given the adjuvanted or the unadjuvanted vaccines (*p* < 0.0001 and *p* < 0.001 respectively) (Appendix A). Similarly, down regulation of γC on ST2/IL-33R^+^ ILC2 was also observed in STAT6^−/−^ mice compared to WT BALB/c given the unadjuvanted vaccine (Appendix A).

Similar to ST2/IL-33R^+^ ILC2, an elevated number of NKp46^−^ ILC1/ILC3 were found to express IL-13Rα2 in the FPV-HIV-IL-4R antagonist adjuvanted vaccine group compared to both WT BALB/c and STAT6^−/−^ mice given unadjuvanted FPV-HIV vaccination (*p* < 0.01) (Figure 3a). Interestingly, STAT6^−/−^ mice given unadjuvanted FPV-HIV vaccine showed down-regulation of the IL-13Rα1 on NKp46^−^ ILC1/ILC3 compared to WT BALB/c mice given the unadjuvanted FPV-HIV or FPV-HIV-IL-4R antagonist adjuvanted vaccines (Figure 3b). Moreover, transient inhibition of STAT6 signaling (WT BALB/c given FPV-HIV-IL-4R antagonist) down-regulated the expression of IL-4Rα and γC (type I IL-4 receptor complex) on NKp46^−^ ILC1/ILC3s (*p* < 0.05) (Figure 3c,d). In the context of IL-4/IL-13 receptor densities, STAT6^−/−^ mice given the unadjuvanted vaccine showed significant downregulation of IL-13Rα2 on NKp46^−^ ILC1/ILC3 compared to the other vaccine groups tested (Appendix A). Interestingly, IL-13Rα1 or IL-4Rα was not modulated on NKp46^−^ ILC1/ILC3 under transient versus permanent blockage of STAT6 (Appendix A), although STAT6 inhibition resulted in down-regulation of γC expression (Appendix A). (For data presented in the form of percentage of CD45^+^ cells, please see Appendix A).

### 3.3. Following FPV-HIV-IL-13Rα2 Adjuvanted Vaccination, IL-13Rα2 Was Not Regulated on ST2/IL-33R^+^ ILC2 or NKp46^−^ ILC1/ILC3, Unlike IL-13^−/−^ Given FPV-HIV Vaccination

Next, IL-4/IL-13 receptor expression on ST2/IL-33R^+^ lung ILC2 was examined following vaccination under transient versus permanent IL-13 inhibition (WT BALB/C given FPV-HIV-IL-13Rα2 vs. IL-13^−/−^ given FPV-HIV). Although expression of IL-13Rα2 on ST2/IL-33R^+^ ILC2 was not modulated, IL-13Rα1 expression was significantly down regulated in IL-13^−/−^ mice given the unadjuvanted vaccine (*p* < 0.05) (Figure 4a,b). Interestingly, an elevated number of WT BALB/c ST2/IL-33R^+^ ILC2 were found to express both γC and IL-4Rα following FPV-HIV-IL-13Rα2 adjuvanted vaccination, unlike IL-13^−/−^ mice given unadjuvanted FPV-HIV vaccine (*p* < 0.001 and *p* < 0.0001, respectively) (Figure 4c,d). Furthermore, compare to the WT BALB/c mice given the unadjuvanted vaccine, WT mice that received the FPV-HIV-IL-13Rα2 adjuvanted vaccine showed significant regulation of type I IL-4 receptor complex (IL-4Rα/γC) on ST2/IL-33R^+^ ILC2 (Figure 4c,d). In the context of IL-4/IL-13 receptor densities, a significantly reduced IL-4Rα/γC expression was observed on IL-13^−/−^ ST2/IL-33R^+^ ILC2 given the unadjuvanted vaccine (Appendix A). Interestingly, IL-13Rα1 and IL-13Rα2 densities were not regulated in any of the vaccine groups tested (Appendix A).

Furthermore, a significantly elevated number of IL-13^−/−^ NKp46^−^ ILC1/ILC3s were found to express IL-13Rα2 given the unadjuvanted vaccine compared to WT BALB/c mice given the unadjuvanted or adjuvanted vaccines (*p* < 0.001), where the latter two groups showed very similar IL-13Rα2 expression profiles (Figure 5a). However, surprisingly, transient inhibition of IL-13 showed an elevated number of NKp46^−^ ILC1/ILC3 expressing IL-13Rα1 compared to WT BALB/c and IL-13^−/−^ mice given the unadjuvanted vaccine (*p* < 0.01) (Figure 5b). In contrast, an elevated number of NKp46^−^ ILC1/ILC3s obtained from IL-13^−/−^ given the unadjuvanted vaccines were also found to express IL-4Rα (*p* < 0.05) (Figure 5c), but not γC (Figure 5d). In the transient versus permanent IL-13 inhibitory vaccination conditions, IL-13Rα2, IL-4Rα, and γC, but not IL-13Rα1 densities, were differentially regulated on NKp46^−^ ILC1/ILC3 (Appendix A). (For data presented in the form of percentage of CD45^+^ cells, please see Appendix A).

### 3.4. Vaccination under Transient or Permanent Inhibition of IL-13 or STAT6 Signaling Did Not Modulate IL-13Rα2 Expression on NKp46^+^ ILC1/ILC3, Unlike NKp46^−^ ILC1/ILC3

When IL-4/IL-13 receptors on NKp46^+^ ILC1/ILC3 were accessed under transient versus permanent inhibition of IL-13 or STAT6 signaling, surprisingly there were no major differences in the number of NKp46^+^ ILC1/ILC3 expressing IL-13Rα2 (even though the number of cells expressing the receptor was lower in the KO mice compared to the BALB/c given the unadjuvanted vaccine, *p* < 0.5) (Figure 6a). In contrast, the number of NKp46^+^ ILC1/ILC3 expressing γC, IL-4Rα, and IL-13Rα1 (Figure 6b,c) was found to be significantly lower under permanent STAT6 or IL-13 inhibition (KO mice) compared to transient inhibition. Interestingly, both transient blockage of STAT6 (FPV-HIV-IL-4R antagonist vaccination of wild type mice) and IL-13 (FPV-HIV-IL-13Rα2 vaccination of wild type mice) showed a significantly elevated number of NKp46^+^ ILC1/ILC3 expressing γC, IL-4Rα, and IL-13Rα1 compared to BALB/c mice given the unadjuvanted FPV-HIV vaccine (Figure 6b,c). (For data presented in the form of percentage of CD45^+^ cells, please see Appendix A).

### 3.5. rFPV and rVV Vaccinated Lung cDCs and pDC Exhibited Uniquely Differential IL-4/IL-13 Receptor Expression Profiles 24–72 h Post-Delivery

We have previously shown that the nature and replication status of a viral vector can significantly alter the ILC2-derived IL-13 level at the vaccination and can modulate lung DC recruitment [30]. Moreover, a low IL-13 environment at the vaccination site can recruit enhanced cDC leading to CD8^+^ T cells of higher avidity [31,32], whist IL-13 was necessary for effective antibody differentiation [33,51]. Knowing that pDCs play a role in effective antibody differentiation [52,53], in this study IL-4/IL-13 receptor regulation on cDCs and pDCs were also assessed 24–72 h post rFPV and rVV vaccination (high and low IL-13 conditions), as described in Appendix A. The results indicated that compared to rFPV, known to induce low ILC2-derived IL-13 [24,30], rVV induced considerably elevated ILC2-derived IL-13 at the lung mucosae by an ST2/IL-33R^−^ ILC subset, 24 h post vaccination (*p* < 0.0001) (Figure 7a). There was significant regulation of the different IL-4/IL-13 receptors on both cDC and pDC at early stages of vaccination. Interestingly, although the percentage of cDCs expressing IL-13Rα2 was much greater 24–48 h (90%) compared to 72 h post rFPV delivery (~80%) (*p* < 0.0001) (Figure 7b), the IL-4Rα and IL-13Rα1 on cDC were significantly up-regulated only after 48 h (24 vs. 48 h and 24 vs. 72 h *p* < 0.0001) (Figure 7b). In contrast, post rVV vaccination, significantly elevated and sustained IL-13Rα2 expression (99%) was detected throughout the time course (Figure 7c), whilst the IL-13Rα1/IL-4Rα expression trends were very similar to rFPV vaccination (Figure 7c). Notably, at these time points γC receptor, which forms the IL-4 type I receptor complex (IL-4Rα and γC), was not expressed or regulated at 72 h post vaccination (Appendix A).

IL-13Rα2 densities, 24 to 72 h post rFPV vaccination, were down regulated (Appendix A), whilst the opposite was observed with IL-13Rα1 and IL-4Rα (Appendix A). In contrast, post rVV vaccination down-regulation of both IL-13Rα2 and IL-13Rα1 densities at 48 h (24 vs. 48 h *p* < 0.0001), followed by an up-regulation at 72 h, comparable to 24 h (Appendix A) and a gradual but significant increase in the IL-4Rα densities, were detected over time (24 vs. 48 h *p* = 0.0127, 48 vs. 72 h and 24 vs. 72 h *p* < 0.0001) (Appendix A). Interestingly, on cDC the IL-13Rα2 densities were approximately ten times greater than IL-13Rα1 and IL-4Rα.

The IL-13Rα2 expression on pDCs post rFPV vaccination was found to be in the order of (24 > 48 < 72 h) (24 vs. 48 h and 48 vs. 72 h *p* < 0.0001) (Figure 7d), whereas rVV showed a significant up-regulation of IL-13Rα2, both at 48 and 72 h, compared to 24 h post-delivery (24 < 48 ≤ 72 h) (24 vs. 48 h and 24 vs. 72 h *p* < 0.0001) (Figure 7e). Interestingly, very a low number of rFPV vaccinated pDCs expressed IL-4Rα, IL-13Rα1, and γC at 24 h and 48 h (≥3%), and no detectable expression was observed at 72 h post-delivery (Figure 7d and Appendix A). In contrast, significant up regulations of IL-4Rα and IL-13Rα1 were detected on rVV vaccinated lung pDCs 48 to 72 h post-delivery (20–80%), where a very high proportion of pDCs expressed IL-13Rα1 (24 vs. 48 and 24 vs. 72 h *p* < 0.0001) and IL-4Rα (24 vs. 48 *p* < 0.0001 and 24 vs. 72 h *p* = 0.0003) compared to at 24 h (≥2%) (Figure 7e). However, less than 2% of rVV vaccinated pDCs expressed γC at 24 h, and no detectable expression was found at other time points (Figure 7e and Appendix A).

Post rFPV vaccination, a significant decrease in IL-13Rα2 (24 vs. 48 h *p* = 0.0002; 24 vs. 72 h *p* = 0.0003), IL-13Rα1 (24 vs. 48 h and 24 vs. 72 h *p* = 0.0284), and IL-4Rα densities (24 vs. 48 h and 24 vs. 72 h *p* = 0.0277) was observed over time (Appendix A). In contrast, rVV vaccinated pDCs showed a significantly elevated IL-13Rα2 density at 72 h (24 vs. 72 h *p* < 0.0001), including the IL-13Rα1 (24 vs. 48 h *p* = 0.0002; 24 vs. 72 h *p* < 0.0001) and IL-4Rα densities (24 vs. 48 h and 24 vs. 72 h *p* < 0.0001) (Appendix A). Interestingly, the density of IL-13Rα2 on rVV vaccinated pDC was approximately 10 times greater than that of IL-13Rα1 and IL-4Rα.

## 4. Discussion

The current findings revealed that, 24 h post intranasal viral vector vaccination, the expression of IL-13 and IFN-γ in lung ILC2 and ILC1/ILC3 subsets were differentially regulated under transient versus permanent blockage of IL-13 and STAT6. This was governed by the regulation of IL-4/IL-13 receptors (type I γC/IL-4Rα, type II IL-4Rα/IL-13Rα1 IL-4 receptor complexes, and IL-13Rα2) on ST2/IL-33R^+^ ILC2 and NKp46^−^ ILC1/ILC3. Specifically, unlike the IL-13 inhibitory conditions, the disruption of STAT6 signaling induced differential IL-13Rα2 expression on ST2/IL-33R^+^ ILC2 and NKp46^−^ ILC1/ILC3, alluding to regulation of IL-13 by IL-13Rα2, under certain conditions (Table 1). Animals given the FPV-HIV-IL-4R antagonist vaccination (which has shown low or no ILC2-derived IL-13 expression [24]), induced an elevated number of ST2/IL-33R^+^ ILC2 expressing IL-13Rα2, whilst, STAT6^−/−^ mice given the unadjuvanted vaccine showed elevated ST2/IL-33R^+^ ILC2-derived IL-13 and very low IL-13Rα2 expression (Table 1). Knowing that IL-13Rα2 is the high affinity receptor for IL-13 (works/signals under low IL-13 on lung DCs) [14], these observations jointly implied the possible co-regulation of ILC2-deived IL-13 and IL-13Rα2 at the vaccination site (Figure 8 and Appendix A), unlike IL-13Rα1 (as expression of the latter was similar between the three groups tested (Figure 2)). Moreover, the down-regulation of both IFN-γ and IL-13Rα2 in STAT6^−/−^ NKp46^−^ ILC1/ILC3 given the unadjuvanted vaccine, and the opposing effect (up-regulation of IFN-γ and IL-13Rα2) observed in NKp46^−^ ILC1/ILC3s, when WT BALB/c mice were given the FPV-HIV-IL-4R antagonist adjuvanted vaccine (transient inhibition of STAT6), indicated that at the lung mucosae the IL-13 and IFN-γ balance was likely inter-regulated by different ILCs (Figure 8 and Appendix A). Interestingly, IL-13^−/−^ mice given the FPV-HIV (which has shown low or no ILC2-derived IL-13 expression [24]), showed elevated expression of IFN-γ by NKp46^−^ ILC1/ILC3, whereas BALB/c mice given the FPV-HIV-IL-13Rα2 adjuvanted vaccine showed reduced ST2/IL-33R^+^ ILC2-driven IL-13, as well as NKp46^−^ ILC1/ILC3-driven IFN-γ expression (Table 1). It is now well recognized that IL-13 can modulate IFN-γ expression [54,55], and our recent findings also showed that ST2/IL-33R^+^ ILC2 can express elevated IFN-γR (Appendix A) (Jeason et al. in preparation). Moreover, an inverse relationship between IL-13Rα2 and IFN-γR expression on lung DCs was recently established following FPV-HIV vaccination [14], associated with low ILC2-derived IL-13 [24,30]. Thus, taken together our findings indicated that redundancies built into the immune system employ a range of regulatory mechanisms to control the balance of ST2/IL-33R^+^ ILC2-driven IL-13 and NKp46^−^ ILC1/ILC3-driven IFN-γ at the vaccination site. Specifically, this balance was likely inter-regulated by IL-13Rα2/IFN-γR on ST2/IL-33R^+^ ILC2 and NKp46^−^ ILC1/ILC3 (Figure 8 and Appendix A).

Our HIV prime-boost vaccine studies have demonstrated that IL-13 signaling via an STAT6 independent pathway, most likely IL-13Rα2, was detrimental for effective IgG1 to IgG2a antibody differentiation following viral vector-based vaccination [33,37] (Table 1). Specifically, IL-4R antagonist vaccination (which interrupted the STAT6 signaling) induced low ILC2-derived IL-13 and elevated NKp46^+/−^ ILC1/ILC3-derived IFN-γ expression at the vaccination site 24 h post-delivery [24]. Interestingly, the role of IFN-γ in effective antibody maturation/development has been well-documented [38,39,40,41,42]. Remarkably, studies have also shown that the IL-13Rα2 cytoplasmic domain can bind to IL-4Rα to prevent STAT6 signaling [17], and IL-4Rα/STAT6 signaling can inhibit IFN-γ expression in CD4^+^ T cells [56]. Recently, IL-13 signaling via IL-13Rα2, leading to expression of TGF-β1 in DCs [14], and association of TGF-β as a key regulator of IgG2a antibody induction, have also been reported [57,58,59]. Therefore, in the context of IL-4R antagonist adjuvanted vaccination (transient blockage of IL-13, and STAT6 signaling) the observed IL-4/IL-13 receptor regulation patterns on ILC also suggested a possible autocrine regulation of ILC2-derived IL-13 in the milieu via IL-13Rα2, as well as the activation of IL-13Rα2 on ILC1/ILC3 to regulate the elevated NKp46^−^ ILC1/ILC3-derived IFN-γ expression (Appendix A). In contrast, in STAT6^−/−^ mice given the unadjuvanted vaccine, receptor regulation patterns indicated sequestration of elevated ST2/IL-33R^+^ ILC2-driven IL-13 by NKp46^−^ ILC1/ILC3 IL-13Rα2, preventing IL-13Rα2 signaling and NKp46^−^ ILC1/ILC3-derived IFN-γ expression (Appendix A). This was similar to what was recently reported in DC under different IL-13 conditions [14]. Interestingly, in this study, following rFPV vaccination, enhanced IL-13Rα2 expression on pDC with no significant IL-13Rα1/IL-4Rα regulation 24–72 h post-delivery was also observed, unlike rVV vaccination (high IL-13 conditions) [14]. Knowing the association of pDCs [52,53] in effective antibody maturation/development, collectively, the current findings once again indicated that IL-13Rα2 plays an important role at the first line of defense, not only at the DC [14] but also ILC level to regulate/maintain the IL-13 and IFN-γ balance at the vaccination site (Figure 8), responsible for diverse adaptive immune outcomes (Table 1).

Intriguingly, the regulation of IL-4/IL-13 receptors on lung ILCs was significantly different under STAT6 compared to IL-13 inhibitory conditions. Although 24 h post FPV-HIV-IL-13Rα2 adjuvanted or unadjuvanted vaccination type I (γC/IL-4Rα) IL-4 receptor complex was regulated on WT BALB/c ST2/IL-33R^+^ ILC2, interestingly, IL-13 receptors (IL-13Rα1 or IL-13Rα2) were not regulated. However, ST2/IL-33R^+^ ILC2 obtained from IL-13^−/−^ mice given the unadjuvanted vaccination showed regulation of type I (γC/IL-4Rα) and type II (IL-4Rα/IL-13Rα1) IL-4 receptor complexes, but showed no detectable expression of IL-13Rα2 or IL-4. Moreover, the expression hierarchy of NKp46^−^ ILC1/ILC3-driven IFN-γ in these mice was in the order: BALB/c unadjuvanted > IL-13−/− unadjuvanted > BALB/c IL-13Rα2 adjuvanted vaccinated (Table 1). Therefore, the observed cytokine and receptor expression profiles suggest that, as IL-4/IL-13 have overlapping specificities, under IL-13^−/−^ conditions, IL-4 expressed by other cells at the lung mucosae may interact with type I and/or II IL-4 receptor complexes on ILC2 to compensate for the loss of IL-13 in the milieu to regulate the ILC/cytokine balance at the vaccination site (e.g., IL-13/IFN-γ). The above findings further corroborated our notion that at the early stages of vaccination, IL-13 is likely the master sensor/regulator of lung ST2/IL-33R^+^ ILC2 and NKp46^−^ ILC1/ILC3 activity/function.

On NKp46^+^ ILC1/ILC3s the expression patterns of IL-4/IL-13 receptors were vastly different compared to the other two ILC subsets. The stable IL-13Rα2 expression on NKp46^+^ ILC1/ILC3 under the transient and control vaccination conditions indicated that these cells were not involved in regulation of ILC2 and NKp46^−^ ILCs, but were likely associated with maintenance of IL-13 homeostasis at the lung mucosae, similar to what has been observed in lung DC under high IL-13 conditions [14], and under IL-13 mediated chronic inflammatory conditions [60,61]. Furthermore, the low IFN-γ expression and minimal regulation of IL-13Rα2 at the early stages of intranasal viral vector vaccination was suggestive of the noninvolvement NKp46^+^ ILC1/ILC3s in the regulation of lung ST2/IL-33R^+^ ILC2.

Unlike rFPV associated with low ILC2-derived, rVV vaccination, which induced significantly elevated ILC2-derived IL-13 (the highest compared to all previously tested viral vectors) [30], showed elevated expression of the high affinity IL-13 receptor IL-13Rα2 on lung DCs 24–72 h post-delivery, including significant up-regulation of the low affinity Type II IL-4Rα/IL-13Rα1 complex at 48–72 h. These receptor regulation patterns once again provoked the notion that under high IL-13, IL-13Rα2 likely sequestered excess IL-13 (noting that rVV is a replicating vector), whilst signaling took place via the low affinity IL-4Rα/IL-13Rα1 complex (which works under high IL-13 conditions) (Figure 8). These findings are highly consistent with our recent observations, where low and high IL-13 conditions showed differential regulation of IL-13Rα2 on DC [14]. These uniquely different early events in the innate immune compartment may explain “how and why” (i) in a prime-boost vaccination modality, rFPV prime can generate high avidity T cells, unlike rVV [62]; and (ii) the order of vector delivery significantly impacts vaccine-specific adaptive immune outcomes [63,64]. Moreover, the observed IL-13/IL-13Rα2 regulation patterns on ILC and DC at the vaccination site may explain why a more attenuated and unrelated viral vector to the host may help induce a higher quality vaccine-specific T cell immunity. Specifically, why rFPV and its relative, canarypox virus prime modalities, may have the capacity to induce more effective immune outcomes than other pox viral vectors [34,63,64,65], given that priming creates the initial antigen-specific T cell population, which gets expanded during the booster vaccination [32,33].

## 5. Conclusions

Collectively, our findings reveal that 24 h post intranasal viral vector vaccination, lung ILC2-derived IL-13 is regulated in an autocrine fashion via IL-13Rα2, and that most likely there is inter-regulation of ILC2-derived IL-13 and NKp46^−^ ILC1/ILC3-derived IFN-γ by their respective receptors (IFN-γR and IL-13Rα2) present on ILC (Figure 8). Specifically, at the early stages of viral vector vaccination: (i) IL-13 is likely the master regulator of ILC2 and NKp46^−^ ILC1/ILC3, as well as DC, and responsible for shaping the downstream adaptive immune outcomes (Table 1); and (ii) IL-13Rα2 is the key IL-13 regulator of both lung ILC and DCs. Thus, taken together with our previous findings, we propose that the IL-13Rα2 and IFN-γR receptor regulation process at the ILC and DC level may play an important role in shaping not only the T cell but also B cell immune outcomes (Table 1), in a vaccine vector, adjuvant, and a route dependent manner; which warrants further investigation.

## Figures and Tables

**Figure 1 vaccines-09-00440-f001:**
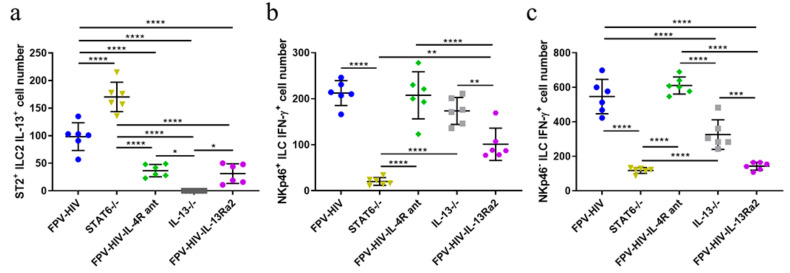
Comparison of ILC2-drived IL-13 and ILC1/ILC3-drived IFN-γ expression following rFPV vaccination under permanent (knock out mice) vs. transient inhibition of IL-13 and STAT6. Color coded vaccine groups represent: blue, WT BALB/c given FPV-HIV (control); yellow, STAT6^−/−^ given FPV-HIV (which represents permanent STAT6 signaling inhibition condition); green, WT BALB/c given FPV-HIV-IL-4R antagonist vaccine (which represents transient IL-4/IL-13/STAT6 signaling inhibition condition); grey, IL-13^−/−^ given FPV-HIV (which represents permanent IL-13 inhibition condition); and purple, WT BALB/c given FPV-HIV-IL-13α2 adjuvanted vaccine (which represents transient IL-13 sequestration/inhibition condition). Graphs represent IL-13 expression by lung IL-33R/ST2^+^ ILC2 (**a**) and IFN-γ expression by lung NKp46^−/+^ ILC1/ILC3 (**b**,**c**) following IL-13^−/−^ and STAT6^−/−^ BALB/c background mice (n = 4) given the control unadjuvanted vaccine (FPV-HIV) compared to BALB/c mice (n = 4) given FPV-HIV unadjuvanted, FPV-HIV-IL-4R antagonist, and FPV-HIV-IL-13Rα2 adjuvanted vaccines. The error bars represent the mean and standard deviation (s.d.). The *p* values were calculated using One-way ANOVA using Tukey’s multiple comparisons test and unpaired *t*-test. * *p* < 0.05, ** *p* < 0.01, *** *p* < 0.001, **** *p* < 0.0001. Experiments were repeated a minimum 3 times. Note that in the figure FPV-HIV-IL-4R ant represents FPV-HIV-IL-4R antagonist.

**Figure 2 vaccines-09-00440-f002:**
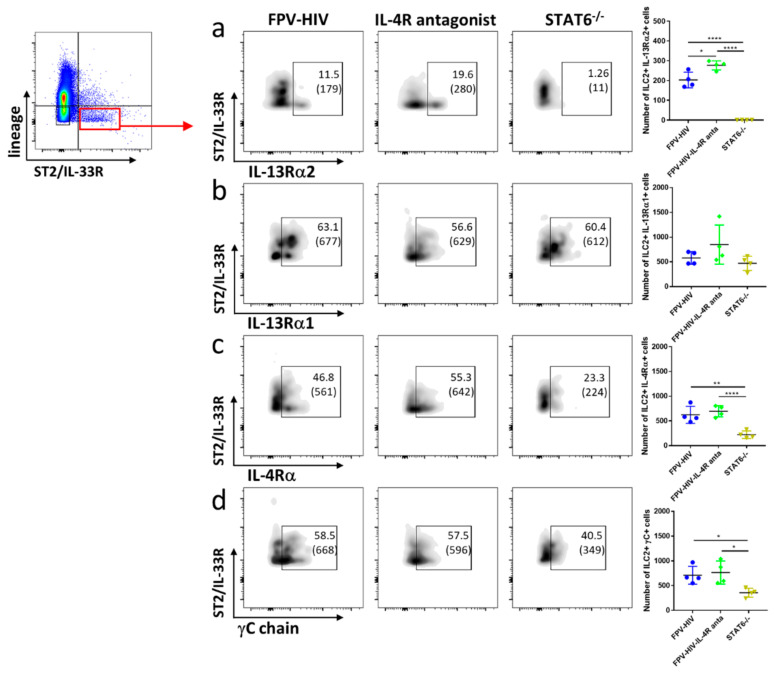
Evaluation of lung lineage^−^ IL-33R/ST2^+^ ILC2 expressing type I (γC/IL-4Rα), type II (IL-4Rα/IL-13Rα1) IL-4 receptor complexes and IL-13Rα2 following vaccination under permanent vs. transient inhibition of STAT6. WT BALB/c and STAT6^−/−^ mice on BALB/c background (n = 4) were immunized intranasally with FPV-HIV-IL-4R antagonist adjuvanted or FPV-HIV unadjuvanted vaccines, respectively. Color coded vaccine groups represent: blue, WT BALB/c given FPV-HIV (control); green, WT BALB/c given FPV-HIV-IL-4R antagonist vaccine (which represents transient IL-4/IL-13/STAT6 signaling inhibition condition); and yellow, STAT6^−/−^ given FPV-HIV (which represents permanent STAT6 signaling inhibition condition). Lung ILC2s were identified as CD45^+^ FSC^low^ SSC^low^ lineage^−^ IL-33R/ST2^+^ cells. The FACS plots in each panel indicate the percentage of ILC2 expressing IL-13Rα2 (**a**), IL-13Rα1 (**b**), IL-4Rα (**c**), and γC chain (**d**) in each vaccinated group. The bracket below the cell percentage indicates the number of cells is each gate. The graph in each panel represents the number of ILC2s expressing the different receptors back calculated to CD45^+^ population as described in the materials and methods (24 h post vaccination). The error bars represent the mean and standard deviation (s.d.). The *p* values were calculated using one-way ANOVA using Tukey’s multiple comparisons test and unpaired *t*-test. * *p* < 0.05, ** *p* < 0.01, **** *p* < 0.0001. Experiments were repeated a minimum of 3 times.

**Figure 3 vaccines-09-00440-f003:**
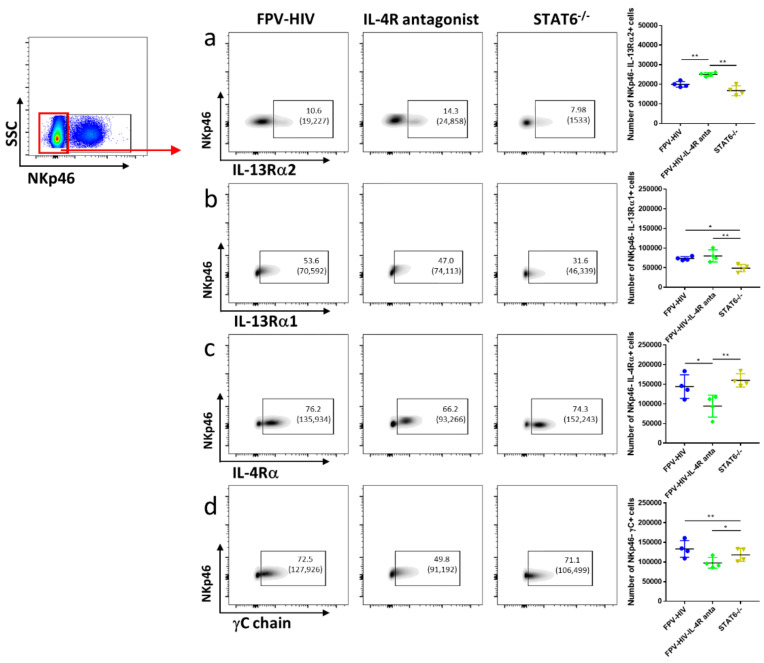
Evaluation of lung lineage^−^ IL-33R/ST2^−^ NKp46^−^ ILC1/ILC3 expressing IL-13Rα2, IL-13Rα1, IL-4Rα, and γC following vaccination under permanent vs. transient inhibition of STAT6. WT BALB/c and STAT6^−/−^ mice on BALB/c background (n = 4) were immunized intranasally with FPV-HIV-IL-4R antagonist adjuvanted or FPV-HIV unadjuvanted vaccines, respectively. Color coded vaccine groups represent: blue, WT BALB/c given FPV-HIV (control); green, WT BALB/c given FPV-HIV-IL-4R antagonist vaccine (which represents transient IL-4/IL-13/STAT6 signaling inhibition condition); and yellow, STAT6^−/−^ given FPV-HIV (which represents permanent STAT6 signaling inhibition condition). Lung NKp46^−^ ILC1/ILC3 were gated as CD45^+^ FSC^low^ SSC^low^ lineage^−^ IL-33R/ST2^−^ NKp46^−^ cells. The FACS plots in each panel indicate the percentage of NKp46^−^ ILC1/ILC3 expressing IL-13Rα2 (**a**), IL-13Rα1 (**b**), IL-4Rα (**c**), and γC chain (**d**). The bracket below the cell percentage indicates the number of cells in each gate. Graph in each panel represents the number of NKp46^−^ ILCI/ILC3s expressing the different receptors back calculated to CD45^+^ population, as described in the materials and methods at (24 h post vaccination). The error bars represent the mean and standard deviation (s.d.). The *p* values were calculated using one-way ANOVA using Tukey’s multiple comparisons test and unpaired *t*-test. * *p* < 0.05, ** *p* < 0.01. Experiments were repeated a minimum of 3 times.

**Figure 4 vaccines-09-00440-f004:**
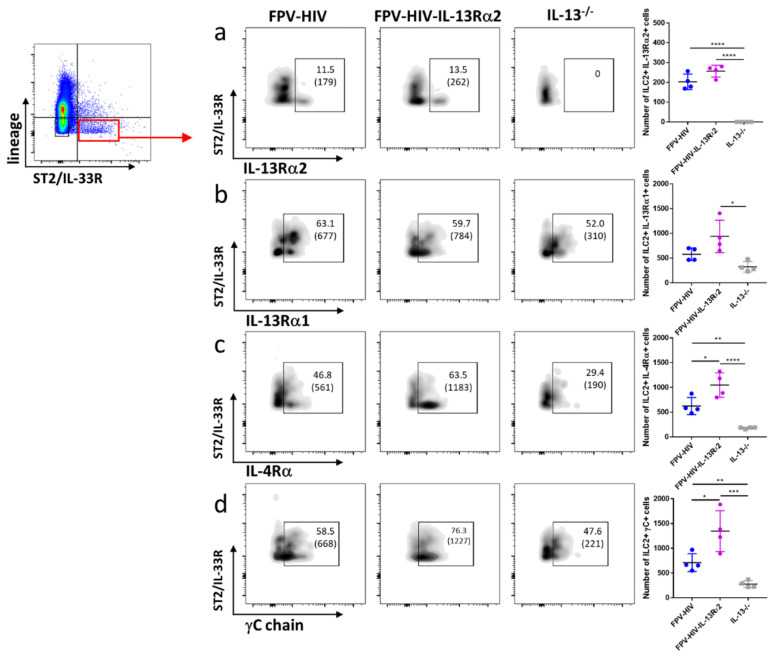
Evaluation of lung lineage^−^ IL-33R/ST2^+^ ILC2 expressing IL-13Rα2, IL-13Rα1, IL-4Rα, and γC following vaccination under permanent vs. transient inhibition of IL-13. WT BALB/c and IL-13^−/−^ mice on BALB/c background (n = 4) were immunized intranasally with FPV-HIV-IL-13Rα2 adjuvanted or FPV-HIV unadjuvanted vaccines. Color coded vaccine groups represent: blue, WT BALB/c given FPV-HIV (control); purple, WT BALB/c given FPV-HIV-IL-13α2 adjuvanted vaccine (which represents transient IL-13 sequestration/inhibition condition); and grey, IL-13^−/−^ given FPV-HIV (which represents permanent IL-13 inhibition condition). Lung ILC2 were gated as CD45^+^ FSC^low^ SSC^low^ lineage^−^ IL-33R/ST2^+^ cells. The FACS plots in each panel indicate the percentage of ILC2s expressing IL-13Rα2 (**a**), IL-13Rα1 (**b**), IL-4Rα (**c**), and γC chain (**d**). The bracket below the cell percentage indicates the number of cells in each gate. The graph in each panel represents the number of ILC2s expressing the different receptors back calculated to CD45^+^ population, as described in the materials and methods (24 h post vaccination). The error bars represent the mean and standard deviation (s.d.). The *p* values were calculated using one-way ANOVA using Tukey’s multiple comparisons test and unpaired *t*-test. * *p* < 0.05, ** *p* < 0.01, *** *p* < 0.001, **** *p* < 0.0001. Experiments were repeated a minimum of 3 times.

**Figure 5 vaccines-09-00440-f005:**
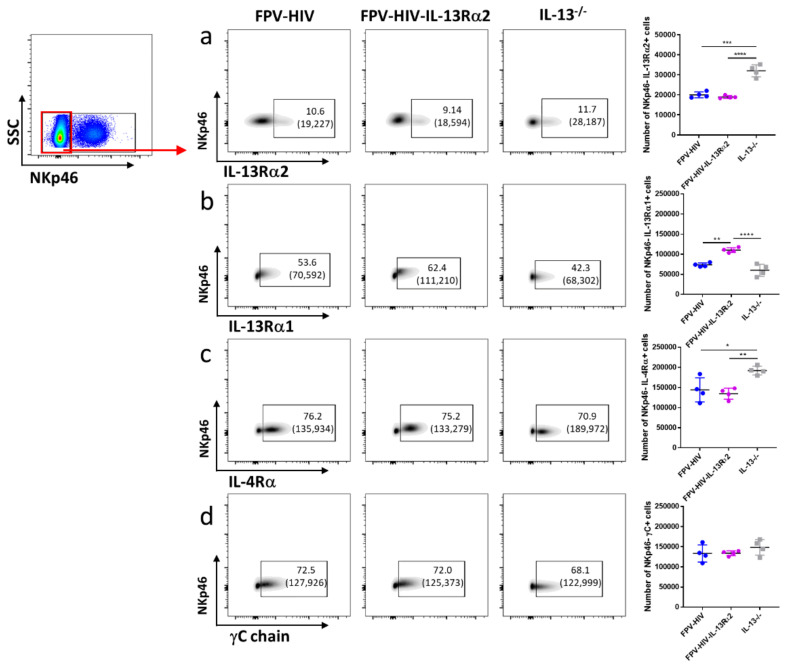
Evaluation of lung lineage^−^ IL-33R/ST2^−^ NKp46^−^ ILC1/ILC3 expressing IL-13R2, IL-13Rα1, IL-4Rα, and αC following vaccination under permanent vs. transient inhibition of IL-13. WT BALB/c and IL-13^−/−^ mice on BALB/c background (each group n = 4) were immunized intranasally with FPV-HIV-IL-13Rα2 adjuvanted or FPV-HIV unadjuvanted vaccines. Color coded vaccine groups represent: blue, WT BALB/c given FPV-HIV (control); purple, WT BALB/c given FPV-HIV-IL-13α2 adjuvanted vaccine (which represents transient IL-13 sequestration/inhibition condition); and grey, IL-13^−/−^ given FPV-HIV (which represents permanent IL-13 inhibition condition). Lung NKp46^−^ ILC1/ILC3 were gated as CD45^+^ FSC^low^ SSC^low^ lineage^−^ IL-33R/ST2^−^ NKp46^−^ cells. The FACS plots in each panel indicate the percentage of NKp46^−^ ILC1/ILC3 expressing IL-13Rα2 (**a**), IL-13Rα1 (**b**), IL-4Rα (**c**), and αC chain (**d**). The bracket below the cell percentage indicates the number of cells in each gate. The graph in each panel represents the number of NKp46^−^ ILC1/ILC3 expressing the different receptors 24 h post vaccination back calculated to CD45^+^ population, as described in the materials and methods. The error bars represent the mean and standard deviation (s.d.). The *p* values were calculated using one-way ANOVA using Tukey’s multiple comparisons test and unpaired *t*-test. * *p* < 0.05, ** *p* < 0.01, *** *p* < 0.001, **** *p* < 0.0001. Experiments were repeated a minimum of 3 times.

**Figure 6 vaccines-09-00440-f006:**
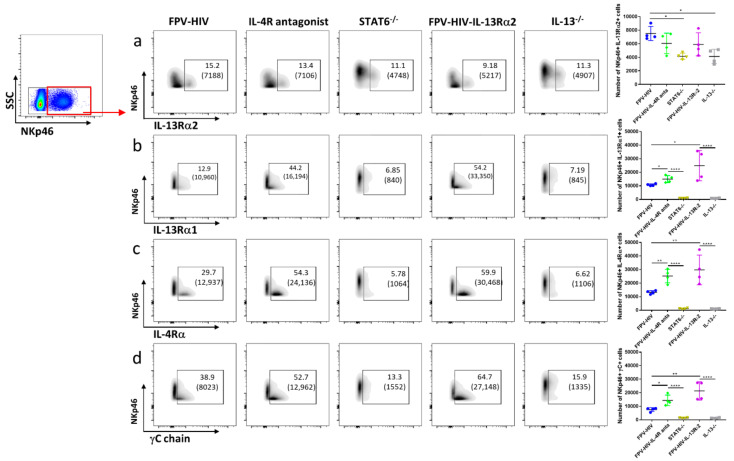
Evaluation of lung lineage^−^ IL-33R/ST2^−^ NKp46^+^ ILC1/ILC3 expressing IL-13Rα2, IL-13Rα1, IL-4Rα, and γC receptors following rFPV vaccination under different inhibitory conditions. WT BALB/c mice (n = 4) were immunized intranasally with FPV-HIV-IL-4R antagonist or FPV-HIV-IL-13Rα2 adjuvanted vaccines. STAT6^−/−^ and IL-13^−/−^ mice on BALB/c background (n = 4) were immunized with FPV-HIV unadjuvanted vaccine. Color coded vaccine groups represent: blue, WT BALB/c given FPV-HIV (control); yellow, STAT6^−/−^ given FPV-HIV (which represents permanent STAT6 signaling inhibition condition); green, WT BALB/c given FPV-HIV-IL-4R antagonist vaccine (which represents transient IL-4/IL-13/STAT6 signaling inhibition condition); grey, IL-13^−/−^ given FPV-HIV (which represents permanent IL-13 inhibition condition); and purple, WT BALB/c given FPV-HIV-IL-13α2 adjuvanted vaccine (which represents transient IL-13 sequestration/inhibition condition). Lung NKp46^+^ ILC1/ILC3 were gated as CD45^+^ FSC^low^ SSC^low^ lineage^−^ IL-33R/ST2^−^ NKp46^+^ cells. The FACS plots in each panel indicate the percentage of NKp46^+^ ILC1/ILC3 expressing IL-13Rα2 (**a**), IL-13Rα1 (**b**), IL-4Rα (**c**), and γC chain (**d**). The brackets below the cell percentage indicate the number of cells in each gate. The graph in each panel represents the number of NKp46^+^ ILC1/ILC3 expressing the different receptors 24 h post vaccination back calculated to CD45^+^ population, as described in the materials and methods. The error bars represent the mean and standard deviation (s.d.). The *p* values were calculated using one-way ANOVA using Tukey’s multiple comparisons test and unpaired *t*-test. * *p* < 0.05, ** *p* < 0.01, **** *p* < 0.0001. Experiments were repeated a minimum of 3 times.

**Figure 7 vaccines-09-00440-f007:**
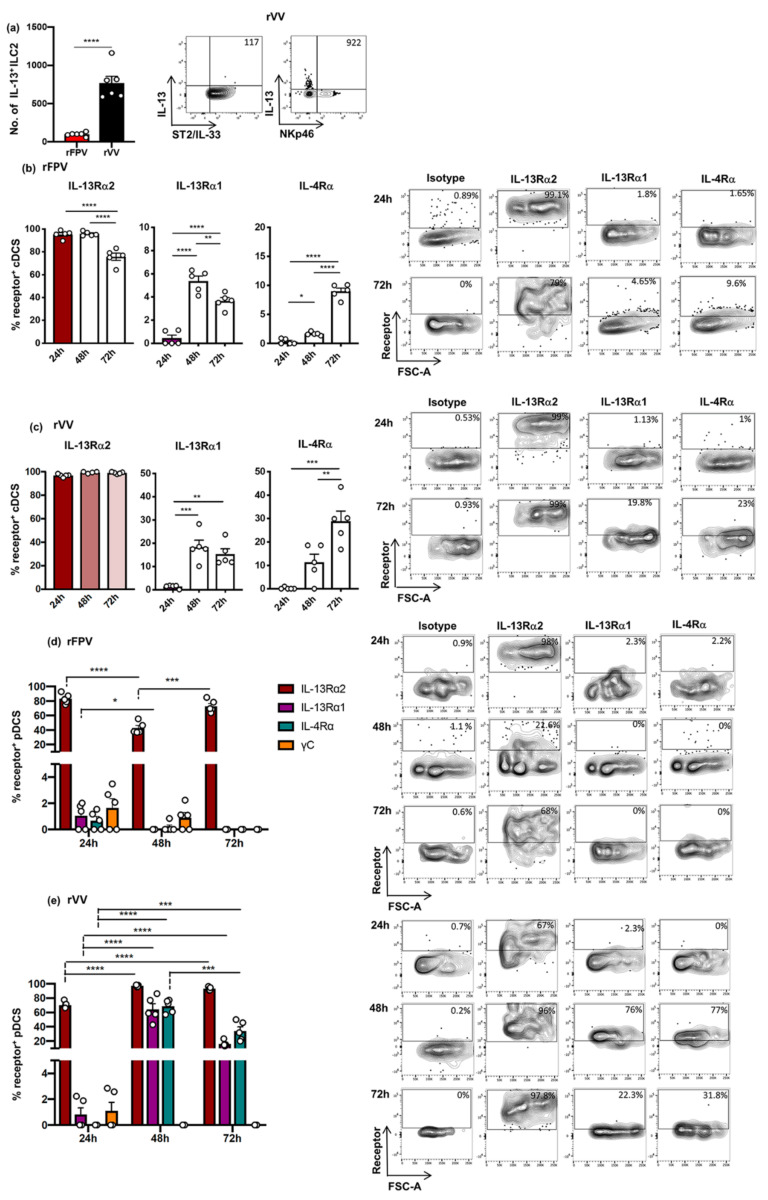
Evaluation of lung ILC2-derived IL-13 expression and lung cDCs and pDCs expressing IL-4/IL-13 receptors, following intranasal rFPV and rVV vaccination. BALB/c mice (n = 6 per group) were immunized i.n. with FPV-HIV or VV-HIV, 24 h post vaccination single cell suspensions from lungs were prepared and stained for lineage^−^ ST2/IL-33R^+^ and lineage^−^ ST2/IL-33^−^ NKp46^−^ ILC2s, and their IL-13 expression was assessed using flow cytometry, as described in the materials and methods. Graphs show (**a**) the number of lineage^−^ ST2/IL-33R^+^ and lineage^−^ ST2/IL-33^−^ NKp46^−^ ILC2s expressing IL-13, 24 h post rFPV and rVV vaccination (left panel). (**a**) Representative FACS plots show the average number of lineage^−^ ST2/IL-33R^+^ and lineage^−^ ST2/IL-33^−^ NKp46^−^ ILC2s expressing IL-13 following rVV vaccination (right panel). BALB/c lungs (n = 5 per vaccine group) were harvested at 24 h, 48 h, or 72 h post rFPV or rVV delivery. Single cell suspensions were prepared and stained for MHC-II^+^ CD11c^+^ CD11b^+^ CD103^−^ cDCs and IL-4/IL-13 receptors and the expression on lung cDCs was assessed using flow cytometry, as described in the materials and methods. Bar graphs (left panel) and representative flow cytometry plots (right panel) show IL-13Rα2, IL-13Rα1, and IL-4Rα expression following vaccination with (**b**) rFPV and (**c**) rVV vaccination. Another set of single cell suspensions from the same vaccinated animals were prepared and stained for IL-4/IL-13 receptors on lung MHC-II^+^ CD11c^+^ CD11b^−^ B220^+^ pDCs. Bar graphs (left panel) and representative flow cytometry plots (right panel) show IL-13Rα2, IL-13Rα1, IL-4Rα, and γC expression at 24 h, 48 h, and 72 h post (**d**) rFPV and (**e**) rVV vaccination. Error bars represent standard error of mean (SEM), and *p* values were calculated using one-way ANOVA followed by Tukey’s multiple comparison test (black lines) and unpaired non-parametric student’s *t* test (grey lines). * *p* < 0.05, ** *p* < 0.01, *** *p* < 0.001, **** *p* < 0.0001. Experiments with each vector were repeated minimum a 2–3 times.

**Figure 8 vaccines-09-00440-f008:**
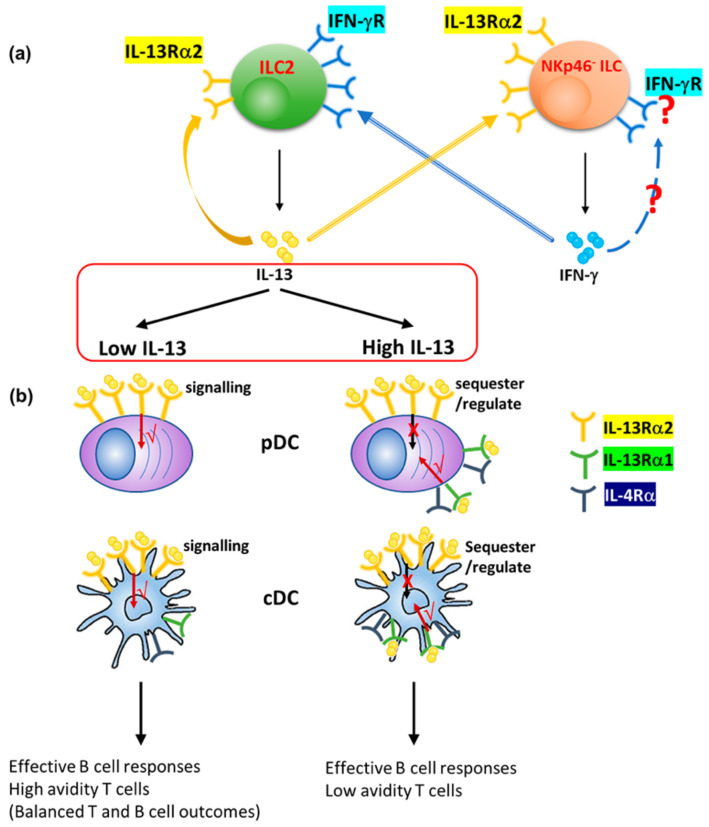
Schematic diagram showing the proposed co-regulation of ILC2-derived IL-13, inter-regulation of ILC2 and ILC1/ILC3 and DC under high and low IL-13. (**a**) At 24 h post viral FPV-HIV vaccination, lung IL-33R/ST2^−^ ILC2-derived IL-13 and IL-13Rα2 were co-regulated. Moreover, the ILC2-derived IL-13 and NKp46^−^ ILC1/ILC3-derived IFN-γ were also inter-regulated by their respective receptors (IFN-γR and IL-13Rα2) present on these ILCs. Taken together our findings indicate that at the early stages of vaccination, IL-13 is the master regulator of different ILC subsets. However, whether there is regulation of IFN-γ via IFN-γR on ILC1/ILC3 warrants further investigation. Moreover, see Appendix A for STAT6 and IL-13 transient versus permanent inhibition scenarios. (**b**) Under low (rFPV) and high (rVV) IL-13 conditions DCs are differentially regulated. Under low IL-13 conditions, IL-13 signals via the high affinity receptor IL-13Rα2 on pDCs and cDCs, inducing effective B cell (e.g., differentiated antibody responses) as well as high avidity/polyfunctional T cells responses [32,33,36,37]. Under high IL-13 conditions, IL-13 signals via the low affinity Type II IL-4 receptor IL-13Rα1/IL-4Rα complex, on pDCs and cDCs, while IL-13Rα2 sequesters/regulates excess IL-13 at the vaccination site maintaining homeostasis, resulting in effective B cell but not T cell outcomes.

**Table 1 vaccines-09-00440-t001:** Summary of ILC-derived cytokine and IL-13Rα2 expression on ILC 24 h post intranasal rFPV vaccination under different IL-4/IL-13 signaling conditions and resulting adaptive immune outcomes *.

Condition	IL-13 (by ILC2)	IL-13Rα2 (ILC2)	IL-13Rα2 (NKp46^−^ ILC)	IL-13Rα2 (NKp46^+^ ILC)	STAT6 Signalling	IFN-γ Level (NKp46^−^ ILC)	IFN-γ Level (NKp46^+^ ILC)	T Cell Avidity *	Antibody Differentiation *
Control	+++	+++	+++	+++	√	+++	+++	+++	++++
◊IL-4R antagonist	±	+++++	+++++	+++	⨂	+++++	+++	+++++	++++
◊STAT6^−/−^	+++++	+	+++	++	⨂	+	+	++	+++++
IL-13Rα2 vaccine	±	+++	+++	+++	√	+	+	+++++	±
IL-13^−/−^	-	-	+++++	++	√	++	+++	+++++	+

Symbols represent: - absent or below detection, ± week or very low, + low, ++ medium, +++ moderate, ++++ high, +++++ very high, √ signaling present, ⨂ signaling absent (permanently or transiently). * Note that the adaptive immune outcomes were published in the following articles [32,33,34,35,36,37]. ◊IL-4R antagonist prime-boost vaccination induced high avidity in both effector and memory T cells (outcomes were confirmed both in mice and macaques [33,34,35]) together with elevated IgG1/IgG2a antibodies, whereas STAT6^−/−^ mice given unadjuvanted strategy generated high avidity memory T cells (not effector) and elevated IgG2a (very low IgG), compared to the WT BALB/c given the unadjuvanted strategy [33,36,37].

## Data Availability

The authors declare that all data supporting the findings of this study are available within the paper and Appendix A.

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
