# Peer review of "IL-13Rα2 Regulates the IL-13/IFN-γ Balance during Innate Lymphoid Cell and Dendritic Cell Responses to Pox Viral Vector-Based Vaccination"

_vaccines, 2021, doi:10.3390/vaccines9050440_

Round 1

Reviewer 1 Report

In this elegant manuscript, the authors demonstrated that the manipulation of IL-13 and STAT6 signaling at the vaccination site can lead to different innate lymphoid cell (ILC)/dendritic cell (DC). The main conclusions can be summarized in these two points: i) IL-13 is likely the master regulator of ILC2, 602 NKp46- ILC1/ILC3 as well as DC, responsible for shaping the downstream adaptive 603 immune outcomes and ii) IL-13Rα2 is the key IL-13 regulator of both lung ILC and DCs.

Overall, this is an interesting study. The data presented are robust and of excellent quality.

- Please avoid acronyms in the title (ILC  and DC)

Reviewer 2 Report

The manuscript talks about how IL-13-mediated signaling can influence the regulation of different ILC/DC subsets. Overall, the findings support the notion that IL-13Ra2 regulates the IL-13 levels in ILC2/DC in an autocrine manner which dictates the outcome of downstream adaptive immune responses. 

I have a few minor suggestions which may improve the readability of the manuscript:

  • Line 56: Please define the acronym ILC2
  • Line 57: Please define the acronym DC
  • I think including some background information on innate lymphoid cell subsets will be very helpful for the readers. 
  • For most of the results discussed, it would be very helpful to have a summary at the end. 

Reviewer 3 Report

In this manuscript, Li and coworkers show that transient (using adjuvanted vaccines) or permanent (using KO mice) inhibition of IL13 or STAT6 signalling can lead to different ILC-derived IL-13 and IFN-γ profiles and differential IL-13Rα2 and IL-4 receptor expression. The authors suggest the presence of inter-regulation between ILC2 and ILC1/ILC3 subsets driven by the production of IL13, IL-13Rα2 and IFN-γ and that this regulation is important in shaping T-cells and B-cells mediated immune outcomes.

The topic is of high interest and the data generated by the authors are compelling. However, in the current form the results are difficult to understand and thus appreciate.

  • Improve readability of the whole manuscript.
  • In figures 1-6 the authors reported the absolute number of cells. Even if the authors are acquiring a fix number of cells it is not clear if the authors are counting events or living cells. This type of analysis make sense in the blood, but in the lungs where cell dissociation plays an important role in the number and quality of cells, it would be better showing percentage of cells out of CD45 cells.

Round 2

Reviewer 3 Report

In the revised version of the manuscript, the authors included a paragraph describing innate lymphoid cells (ILCs) and a figure showing data generated by using flow cytometry showing percentage values. The authors did not improve readability of the manuscript.

The authors should rewrite the manuscript, for example using shorter sentences, reducing redundant phrases (e.g. “…induced reduction…” should be written as “..reduced..”, etc), and avoiding phrases with multiple interpretations (e.g. “ILC-derived IL13 modulates DC activity”, activates DC migration? Induces DC apoptosis? enhances DC ag presentation?). Furthermore, the methods section does not report enough information to describe how the experiments were performed, for example what strategies were used to block IL4 signaling transiently, how were viral particles produced?
